

# Characteristics of planktonic and sediment bacterial communities in a heavily polluted urban river

Heqing Huang[1], Jianhui Liu[1], Fanghui Zhang[1], Kangwen Zhu[2], Chunhua Yang[1], Qiujie Xiang[1] and Bo Lei[1]

[1] Chongqing Academy of Ecology and Environmental Sciences, Chongqing, China
[2] College of Resources and Environment, Southwest University, Chongqing, China

## ABSTRACT

Urban rivers represent a unique ecosystem in which pollution occurs regularly, altering the biogeochemical characteristics of waterbodies and sediments. However, little is presently known about the spatiotemporal patterns of planktonic and sediment bacterial community diversities and compositions in urban rivers. Herein, Illumina MiSeq high-throughput sequencing was performed to reveal the spatiotemporal dynamics of bacterial populations in Liangtan River, a heavily polluted urban river in Chongqing City (China). The results showed the richness and diversity of sediment bacteria were significantly higher than those of planktonic bacteria, whereas a strong overlap (46.7%) in OTUs was identified between water and sediment samples. Bacterial community composition remarkably differed in waters and sediments. Planktonic bacterial communities were dominated by *Proteobacteria, Bacteroidetes, Cyanobacteria* and *Actinobacteria*, while sediment bacterial communities mainly included *Proteobacteria, Actinobacteria, Chloroflexi* and *Bacteroidetes*. Additionally, several taxonomic groups of potential bacterial pathogens showed an increasing trend in water and sediment samples from residential and industrial areas (RI). Variation partition analysis (VPA) indicated that temperature and nutrient were identified as the main drivers determining the planktonic and sediment bacterial assemblages. These results highlight that bacterial communities in the polluted urban river exhibit spatiotemporal variation due to the combined influence of environmental factors associated with sewage discharge and hydropower dams.

## INTRODUCTION

Bacteria are the important components of river ecosystems and play a vital role in biogeochemical processes of nutrients and biotransformation of organic matter (*Madsen, 2002*; *Wang et al., 2017*). Bacteria communities are extremely sensitive to environmental perturbations and serve as potential integrative indicators for measuring the stability of river ecosystems (*Marshall et al., 2008*; *Zhang et al., 2016*). Therefore, it is particularly

Corresponding author
Bo Lei, 286564920@qq.com

critical to unravel the bacterial community diversity and structure to understand the responses of bacteria to elevated environmental pressures in urban rivers (*Crump et al., 2007*; *Quan et al., 2010*).

Several previous studies have documented the shift in planktonic bacterial community diversity and composition can be influenced by water physicochemical properties including organic matter, nitrogen, phosphorus, pH, and temperature (*Esteves, Lôbo & Hilsdorf, 2015*; *Fortunato et al., 2013*; *Liu et al., 2012*; *Adhikari et al., 2019*; *Yang et al., 2019*; *Zhang et al., 2019a*). Some of these influential factors might collectively regulate planktonic bacterial assemblages in river ecosystems. Moreover, the distribution of sediment bacterial community in river ecosystems has received increasing attention. A number of determined environmental factors have been proposed to play a pivotal role in river sediment bacterial communities including organic matter, nitrogen, phosphorus, pH and heavy metal (*Bouskill et al., 2009*; *Peng et al., 2019*; *Su et al., 2019*; *Xia et al., 2014*; *Xie et al., 2016*). However, water column and sediments are two different habitats in river ecosystems, characterized by distinctive properties, which result in profound dissimilarity of planktonic and sediment bacterial communities (*Liu et al., 2018*; *Mao et al., 2019*). Thus, there is a need for comparative analysis of planktonic and sediment bacterial communities in river ecosystems and their driving factors, given the heterogeneity of habitats in space and time.

Compared with natural aquatic ecosystems, urban rivers represent a heterogeneous ecosystem that is mainly influenced by the surrounding terrestrial environment (*Kaestli et al., 2017*). Sewage from the residential and industrial areas containing excessive pollutants notably alters the basic parameters of water and sediments, such as dissolved oxygen, nitrogen, temperature, and pH, which could further reshape the bacterial community structure and diversity (*Drury, Rosi-Marshall & Kelly, 2013*; *Esteves, Lôbo & Hilsdorf, 2015*; *Mark Ibekwe, Ma & Murinda, 2016*). Sewage also introduces microbiological contamination (i.e., bacterial pathogens) to the urban aquatic ecosystems which can impact the composition and function of bacterial community (*Drury, Rosi-Marshall & Kelly, 2013*; *Mark Ibekwe, Ma & Murinda, 2016*). Additionally, construction of dams considerably modifies water discharge, regulate flow circulation and disturb nutrient transport by retaining suspended materials, thereby altering the hydrodynamic condition of rivers and the distribution of bacterial community (*Dai & Liu, 2013*; *Yan et al., 2015*). Substantial differences in bacterial community composition have been observed between the upstream and downstream of dams (*Ruiz-Gonzalez et al., 2013*; *Liu et al., 2018*). However, the joint effects of sewage discharge and dam construction on bacterial communities in a certain urban river remain inadequately understood.

Considering this background, the biogeography of bacterial communities was studied in Liangtan River, a heavily polluted urban river in Chongqing City, China. Specifically, the aim of this current study was to investigate the spatiotemporal dynamics of both planktonic and sediment bacterial populations in the urban river and their responses to dam construction and sewage discharge. Our findings will improve the understanding of multiple environmental gradients on bacterial communities and provide an ecological reference for assessing urban aquatic ecosystems.
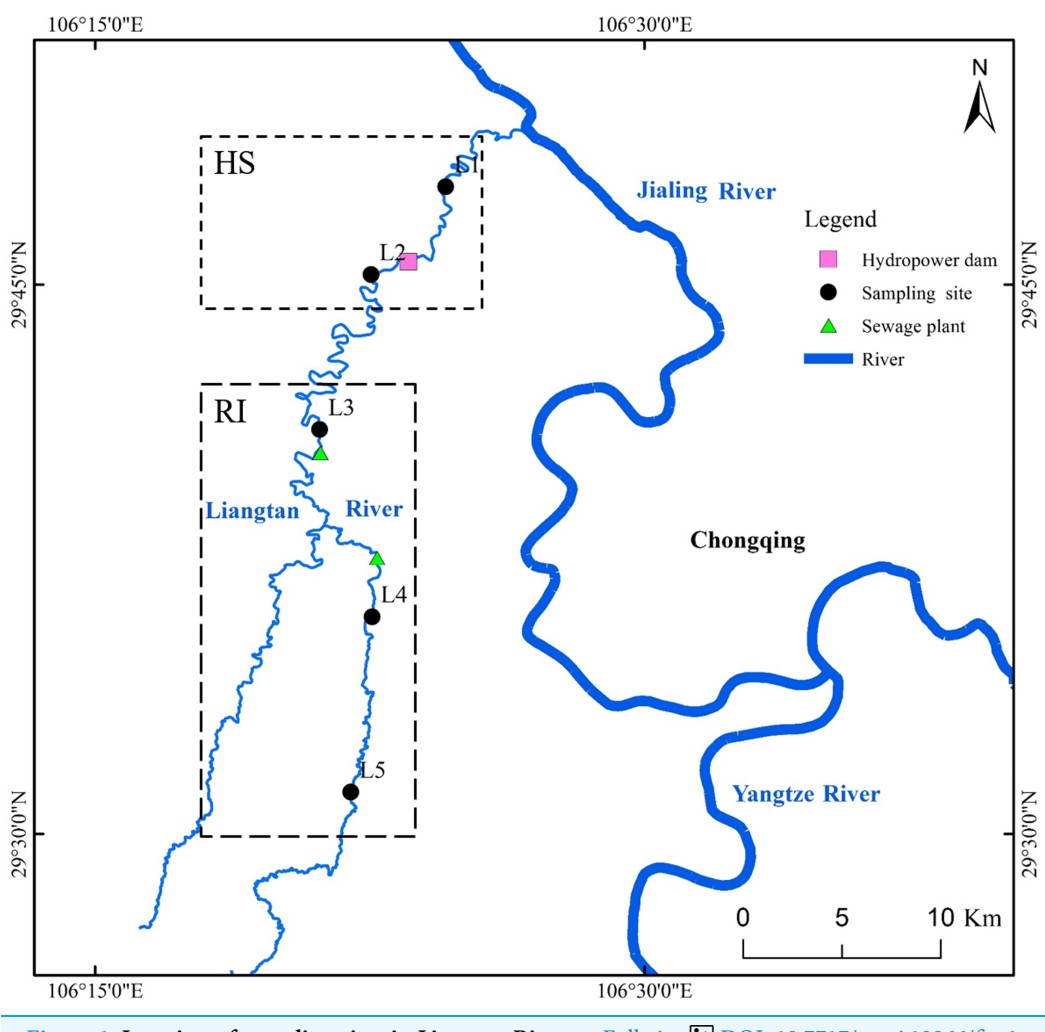

**Figure 1 Location of sampling sites in Liangtan River.**

## MATERIALS AND METHODS

### Sites description and sample collection

Liangtan River (watershed area of 498 km$^2$ and mainstream length of 88 km), located in Chongqing City (China), is one of major tributaries of Jiangling River. As a typical urban river, Liangtan River is completely surrounded by residential and industrial areas. Excessive anthropogenic input results in the deterioration of this urban river. Based on the Environmental Quality Standards for Surface Water of China, the water quality in all sections of Liangtan River has always been classified as worse than grade 5 (*Ran et al., 2016*). In this study, five different sampling sites were selected from upstream to downstream in Liangtan River (Fig. 1). The distance between two adjacent sampling sites is approximately 8–12 km. Sites L1 and L2 are located downstream and upstream of Xiema hydropower station (named as HS), respectively. Sites L3, L4, and L5 are surrounded by residential and industrial areas (named as RI), where the increasing anthropogenic activities have had a great impact on the health of the river ecosystem. Additionally, Sites

L3 and L4 are adjacent to Xiyong and Tuzhu sewage plants, respectively. The sewage runoff from RI area is mainly composed of domestic pollution such as feces, nitrogen, and phosphorus etc. Triplicate water samples (30 cm depth below water surface, about 10 L) and surface sediments (0–10 cm) were collected from the five sampling sites of Liangtan River in February (winter; the average air temperature is 10–25 °C; the average rainfall is 59.8 mm) and August (summer; the average air temperature is 20–29 °C; the average rainfall is 438.3 mm) in 2019, using plexiglass water sampler and tube core sampler, respectively. These water samples and surface sediments were placed in sterile containers and immediately transported to the laboratory for subsequent chemical and molecular analysis.

The physicochemical properties (pH, temperature, ammonia nitrogen ($NH_4^+$–N), nitrate nitrogen ($NO_3^-$–N), total nitrogen (TN), total phosphorus (TP), and total organic carbon (TOC)) of water and sediment samples were measured following the methods described in literature (*China Environmental Protection Agency, 2002*; *Wang, 2012*) and were shown in Tables S1 and S2, respectively.

## Molecular analyses

In the present study, 0.22-µm pore-size membrane (diameter 50 mm; Millipore) was used to retain water microbial cells. Genomic DNA of water and sediment samples were extracted using E.Z.N.A. Water DNA kit (Omega, Norcross, GA, USA) and Powersoil DNA extraction kit (Mobio Laboratories, Carlsbad, CA, USA), respectively, according to manufacturer instructions. The V3–V4 region of bacterial 16S ribosomal RNA (rRNA) gene was amplified using the primer sets 338F (5′-ACTCCTACGGGAGGCAGCAG-3′)/ 806R (5′-GGACTACHVGGGTWTCTAAT-3′), as previously described (*Xu et al., 2016*). The amplicons from each triplicate sample were pooled in equal amounts and sequenced on an Illumina MiSeq platform at Shanghai Majorbio Bio-pharm Technology Co., Ltd. (China). The obtained raw reads were deposited in the NCBI Sequence Read Archive (SRA) database under accession numbers SRP248929 for water and sediment samples. The reads from the original DNA fragments were merged using FLASH and further processed following the protocol by *Caporaso et al. (2010)*. Chimeric reads were discarded using UCHIME (*Edgar et al., 2011*). Sequences were assigned to their corresponding samples according to the barcode sequence and then quality-trimmed with an average Phred quality score cutoff of 20. After quality filtering and the removal of chimeric sequences, data analyses were performed to standardize the sequencing effort across samples. Chimeric-free sequences sharing ≥97% similarity were grouped into operational taxonomic units (OTUs) and alpha-diversity (Chao1 richness estimator, Shannon diversity index, and Simpson evenness index) was further obtained using the UPARSE pipeline (*Edgar, 2013*). The representative sequences of each OTU were classified with the Ribosomal Database Project (RDP) classifier (*Wang et al., 2007*).

## Statistical analysis

One-way analysis of variance (ANOVA), followed by Fisher's least significant difference (LSD) test, was used to determine significant differences ($P < 0.05$) between alpha-diversity

indices and the relative abundances of potential bacterial pathogens among sample groups. Potential relationships between physicochemical properties and (i) bacterial community richness; (ii) diversity; and (iii) the proportion of the major bacterial groups were examined with Spearman rank correlation analysis using the software SPSS 20.0.

To elucidate the microbial interactions of different environments, molecular ecological networks were constructed in Molecular Ecological Network Analysis pipeline (MENA, http://ieg4.rccc.ou.edu/mena/). Person correlation analysis based on random matrix theory (RMT) was used to construct networks. Each network analysis used bacterial OTUs data from ten samples in every ecosystem. OTUs detected in more than half of the samples were kept for the network construction as default settings in the pipeline to ensure the accuracy of OTUs interrelationships. Global network properties, individual nodes' centrality, and module separation and modularity were calculated. Among these properties, each node is an estimable molecular marker (e.g., OTU) and each edge (e.g., interaction) represents the strength of the correlation between them. Logarithm transitivity was used to infer the stability of the community. The constructed network graphs were visualized by Cytoscape (version 3.7.2) and Gephi software (version 0.9.2).

To examine the difference in the overall community composition between each pair of samples at the OTU level, Beta-diversity using Bray–Curtis dissimilarities (*Bray & Curtis, 1957*) was used to examine the difference in the overall community composition between samples and visualized on a PCoA plot. Permutational multivariate analysis of variance (PerMANOVA) was performed to test for significant differences between sample groups based on Bray–Curtis distance matrices. Multivariate analysis was used to determine which physicochemical properties had the biggest impact on planktonic and sediment bacterial communities. Detrended correspondence analysis (DCA) was first applied to determine the suitable ordination analysis method for exploring the correlations between overall bacterial community composition (OTU level) and the environmental factors. The longest DCA axis had a gradient length less than 3 standard deviation units, so redundancy analysis (RDA) was performed (*Lepš & Šmilauer, 2003*; *Li et al., 2019*). Variance inflation factor (VIF) was used for multicollinearity among environmental variables; variables with VIF > 10 were eliminated from RDA analysis. Variation partition analysis (VPA) was applied for assessing the individual and shared effects of season (temperature and rainfall), nutrient ($NH_4^+$–N, $NO_3^-$–N, TN, TP, and TOC) and geographic distance (latitude and longitude) on planktonic and sediment bacterial community structures. Species data were log transformed and environmental data were z-score standardized prior to beta-diversity calculations. R (version 4.0.2) was used for statistical testing unless otherwise indicated. PCoA, PerMANOVA, DCA, RDA, VIF and VPA were performed using the "vegan" package.

## RESULTS

### Bacterial community richness and diversity

In this study, the obtained high-quality bacterial reads from each sample ranged between 28,133 and 63,295, normalized to 28,133 for the comparison of bacterial community richness and diversity. High Good's coverage indicated that the OTUs of each bacterial

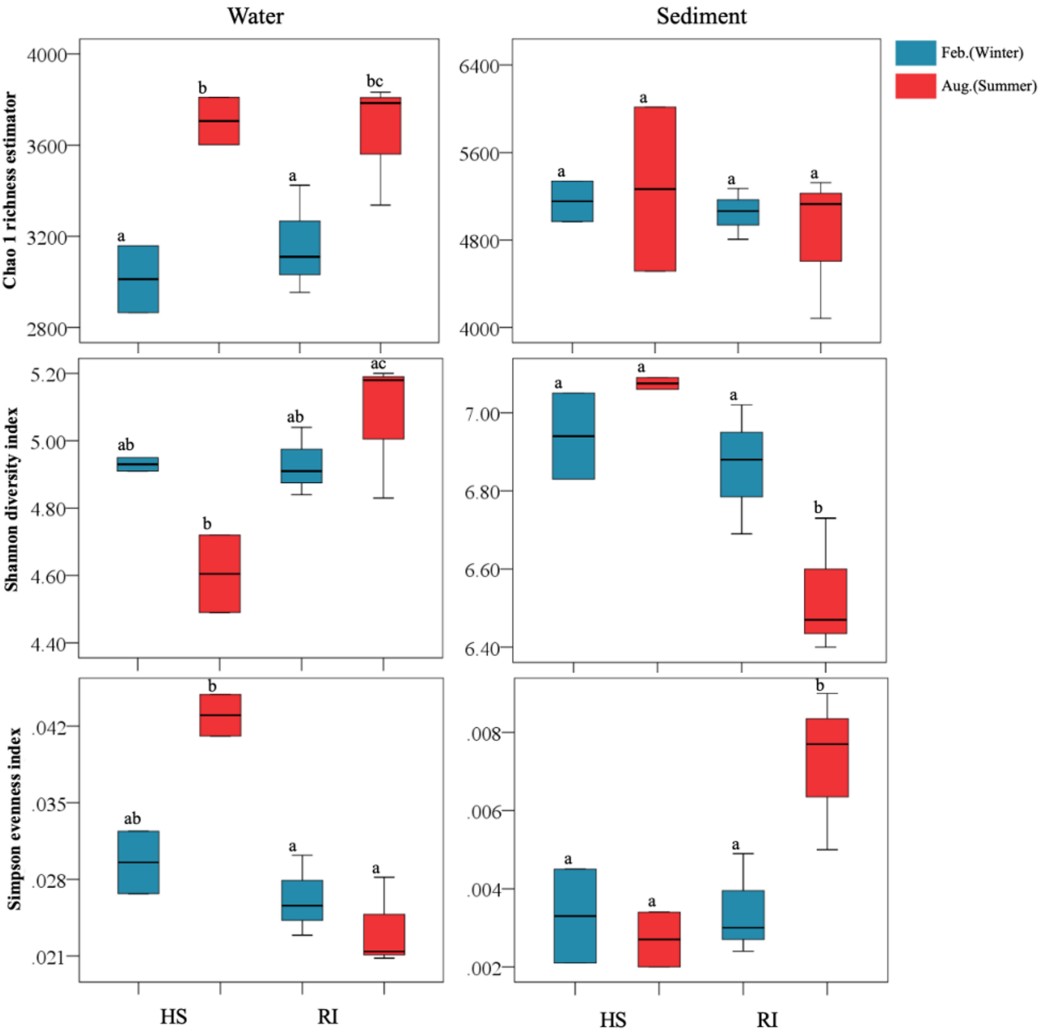

**Figure 2** Comparisons of Chao1 richness estimator, Shannon diversity index, and Simpson evenness index of planktonic and sediment bacterial communities in HS and RI areas. Different lowercase letters indicate significant differences ($P < 0.05$) in mean values.

library had been well captured, with the values ranging from 95.5% to 98.4%. Water bacterial libraries were composed of 1,706–2,783 OTUs, while sediment bacterial libraries comprised 3,488–4,731 OTUs (Table S1). The Chao1 richness estimators of planktonic and sediment bacterial communities were 2,865–3,810 and 4,083–6,015, respectively. At a given sampling site, sediment sample was generally found to have more OTUs and higher Chao1 richness estimator than the corresponding water one. Additionally, a significant increase of planktonic bacterial richness was observed in summer ($P < 0.05$) (Fig. 2), yet the trend of seasonal variation of sediment bacterial richness was not clear. Shannon and Simpson indices of planktonic bacterial communities were 4.49–5.20 and 0.0208–0.0449, while those of sediment bacterial communities were 6.40–7.09 and 0.002–0.009. In summer, Shannon and Simpson indices of both planktonic and sediment bacterial communities in HS area were significantly different from those in RI area
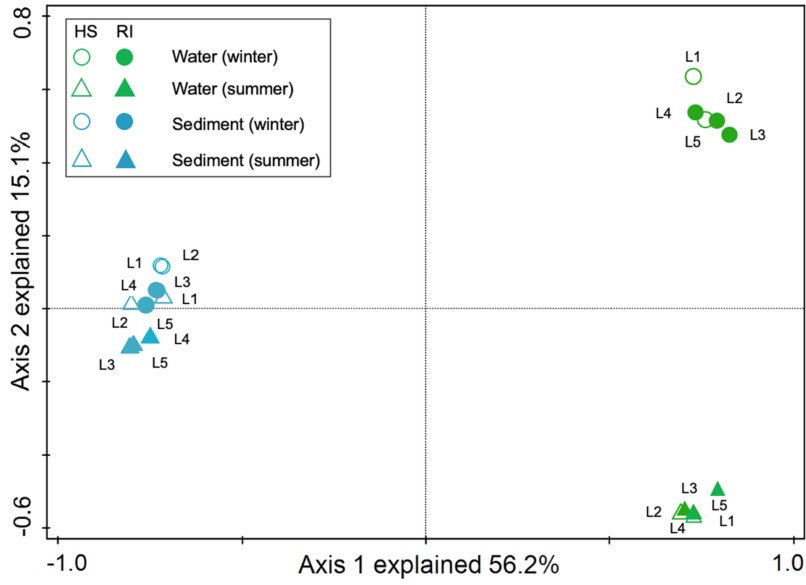

**Figure 3 Principal coordinates analysis (PCoA) for spatiotemporal variations of planktonic and sediment bacterial communities according to Bray-Curtis distancematrices. Green circle symbols represent the February water samples from sites L1 to L5.** Green triangle symbols represent the August water samples from sites L1 to L5. Blue circle symbols represent the February sediment samples from sites L1 to L5. Blue triangle symbols represent the August sediment samples from sites L1 to L5.

($P < 0.05$). However, a significant seasonal difference of sediment bacterial diversity (Shannon and Simpson indices) was only observed in RI area ($P < 0.05$).

In order to explore the interactions between water and sediment bacterial communities, the OTUs were used to construct Microbial Ecological Networks (MENs) (Figs. S1 and S2). A total of 46.7% of OTUs were shared between water and sediment samples. Water samples contained 1,273 unique OTUs that accounted for 14.0% of the total OTUs. The sediment network contained 2,801 edges among 1,746 nodes, which were fewer than the networks of water (9,927 edges among 795 nodes), reflecting a smaller number of co-occurrences in sediment (edge/node ratio = 1.60). The decreases in average connectivity and average clustering coefficient further demonstrated a reduction in network complexity from water to sediment (Table S2). Additionally, 51.3% positive edges and 48.7% negative edges were presented in sediment while 47.5% positive edges and 52.5% negative edges showed in water. These results show that the topological features between water and sediment were distinguishable, and the interactions among the bacterial communities in water were much complex than that in sediment. The results of PCoA analysis revealed a marked discrepancy in the structure of planktonic and sediment bacterial communities (Fig. 3). PerMANOVA confirmed that the bacterial communities in the water were significantly different from those in the sediment ($R^2 = 0.561$, $P = 0.001$) (Table 1). Planktonic bacterial communities for summer samples significantly differed from winter samples ($R^2 = 0.731$, $P = 0.011$), yet the trend for seasonal change of sediment bacterial community structure was not apparent. An obvious and significant difference of sediment bacterial communities was found between HS and RI areas ($R^2 = 0.345$,

**Table 1 Quantitative effects of seasons (winter or summer) and areas (HS or RI) on planktonic and sediment bacterial communities using permutational multivariate analysis of variance (PerMANOVA).**

| Groups | Water | | Sediment | |
|---|---|---|---|---|
| | (Water vs. sediment: $R^2$ = 0.561, $P$ = 0.001) | | | |
| | $R^2$ | $P$ | $R^2$ | $P$ |
| Season (winter vs. summer) | 0.731 | **0.011** | 0.197 | 0.063 |
| Area (HS vs. RI) | 0.120 | 0.496 | 0.345 | **0.006** |
| Season × area | 0.886 | **0.002** | 0.611 | **0.001** |

**Note:**
The data in bold indicate significant differences ($P$ < 0.05).

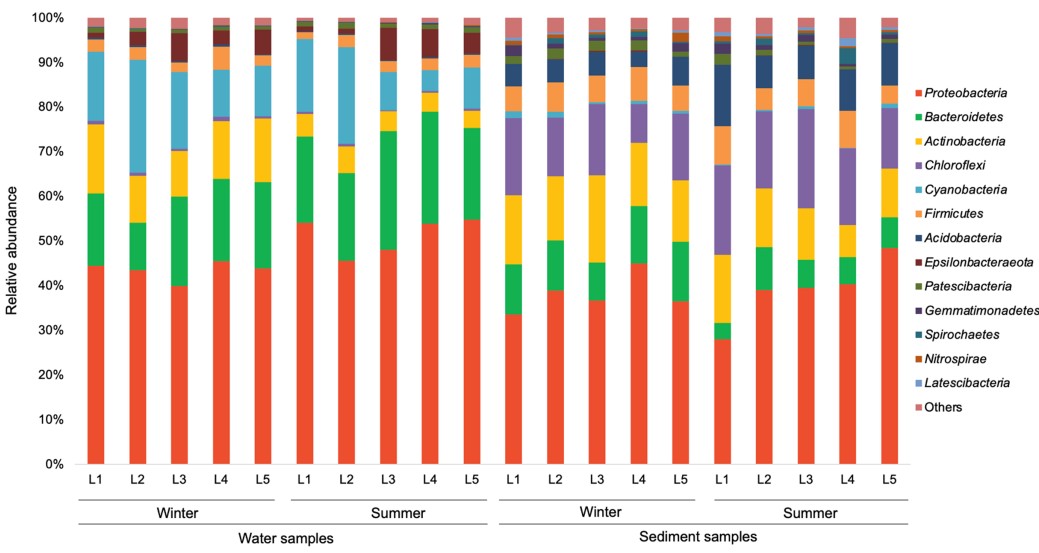

**Figure 4 Relative abundance of dominant bacterial phyla in waters and sediments derived from the sampling sites. Bacterial phyla with relative abundance less than 1% of in each sample are classified as others.**

$P$ = 0.006). Additionally, the interaction effects of seasons (winter or summer) and areas (HS or RI) on planktonic and sediment bacterial communities were statistically significant ($R^2$ = 0.886, $P$ = 0.002; $R^2$ = 0.886, $P$ = 0.002, respectively).

## Bacterial community composition

In the present study, a total of 13 major bacterial phyla (relative abundance >1% in each sample) were identified, including *Proteobacteria*, *Bacteroidetes*, *Actinobacteria*, *Chloroflexi*, *Cyanobacteria*, *Firmicutes*, *Acidobacteria*, *Epsilonbacteraeota*, *Patescibacteria*, *Gemmatimonadetes*, *Spirochaetes*, *Nitrospirae*, and *Latescibacteria* (Fig. 4). *Proteobacteria* showed the dominance in the water samples (39.9–54.8%) and sediment samples (28.0–48.4%), mainly consisting of three classes (*Alphaproteobacteria*, *Gammaproteobacteria*, and *Deltaproteobacteria*) (Fig. 5). *Gammaproteobacteria* predominated proteobacterial classes across all samples, and was relatively higher in water samples. Moreover, winter water samples generally had higher *Gammaproteobacteria*

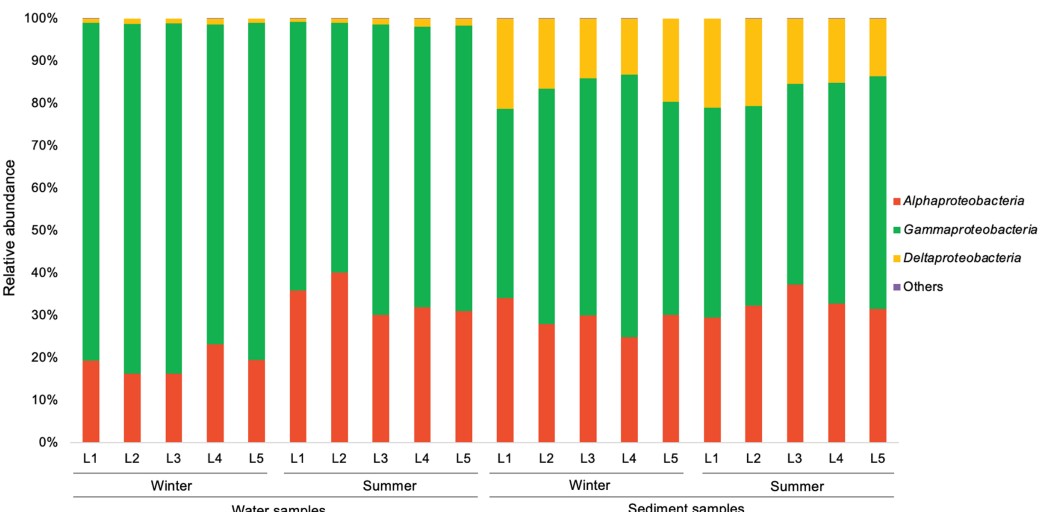

**Figure 5 Relative abundance of proteobacterial classes in waters and sediments derived from the sampling sites. Unclassified Proteobacteria are classified as others.**

proportion than summer samples, yet the seasonal variation of *Gammaproteobacteria* organisms in sediments was not clear. *Alphaproteobacteria* was the second largest proteobacterial class in all samples, but the abundance of planktonic *Alphaproteobacteria* organisms revealed an evident decrease in winter. *Deltaproteobacteria* was also an important proteobacterial member in sediment, but showed lower proportion in water. The second largest bacterial phylum in water samples was *Bacteroidetes* (10.7–26.5%), but its proportion was relatively low in sediment samples (6.1–13.3%). However, *Actinobacteria* and *Firmicutes* were more prevalent in sediment samples (*Actinobacteria* 7.1–19.5%, *Firmicutes* 4.1–8.7%) than those in water samples (*Actinobacteria* 3.9–15.4%, *Firmicutes* 1.5–5.2%). Additionally, *Chloroflexi* and *Acidobacteria* showed relatively high proportion in sediment samples, but became the minor planktonic bacterial groups. In contrast, *Cyanobacteria* and *Epsilonbacteraeota*, as two major components of planktonic bacterial communities, were rarely detected in sediments.

To determine the heterogeneity of potential bacterial pathogens at the genus level among the sample groups, a total of 12 major bacterial genus (relative abundance >0.1% in each sample) were detected, including *Pseudomonas*, *Arcobacter*, *Aeromonas*, *Escherichia-Shigella*, *Legionella*, *Acinetobacter*, *Enterobacter*, *Bacteroides*, *Flavobacterium*, *Mycobacterium*, *Bacillus*, and *Clostridium* (Table S3). Water samples were generally found to have a higher proportion of bacterial pathogens as mentioned above than sediment samples. In either summer or winter, the average relative abundance of potential bacterial pathogens in water and sediment samples from RI area was significantly higher than those from HS areas ($P < 0.05$) (Table 2).

## Influential factors on bacterial communities

Spearman rank correlation analysis indicated that planktonic bacterial community richness (OTU number and Chao1 estimator) and the taxa (*Proteobacteria* and

**Table 2 Comparisons of potential bacterial pathogen sequences at the genus level in waters and sediments derived from HS and RI areas. The data in bold indicate significant differences ($P < 0.05$) in mean values.**

| Season | Area | Water/sediment | Groups | | | | Water/sediment | Area | Season |
|---|---|---|---|---|---|---|---|---|---|
| | | | Winter | | Summer | | | | |
| | | | HS | RI | HS | RI | | | |
| Winter | HS | Water | – | **0.024** | 0.089 | 0.123 | Sediment | HS | Winter |
| | RI | Water | **0.019** | – | **0.002** | 0.228 | Sediment | RI | |
| Summer | HS | Water | 0.148 | 0.217 | – | **0.007** | Sediment | HS | Summer |
| | RI | Water | 0.000 | **0.003** | **0.001** | – | Sediment | RI | |

**Table 3 Spearman rank correlation analysis of water environmental factors with the richness and diversity of planktonic bacterial community or the proportion of the major planktonic bacterial groups. The data in bold indicate significant differences.**

| Parameters | pH | Temperature | $NH_4^+$–N | $NO_3^-$–N | TN | TP | TOC |
|---|---|---|---|---|---|---|---|
| OTUs | 0.012 | **0.669**[a] | −0.103 | −0.067 | −0.382 | **−0.770**[b] | −0.139 |
| Chao1 estimator | 0.012 | **0.693**[a] | −0.236 | 0.067 | −0.236 | **−0.770**[b] | −0.103 |
| Shannon index | 0.255 | −0.264 | **−0.745**[a] | 0.358 | 0.115 | 0.345 | **−0.697**[a] |
| *Proteobacteria* | −0.134 | **0.865**[b] | −0.188 | −0.127 | −0.370 | **−0.673**[a] | −0.115 |
| *Alphaproteobacteria* | −0.158 | **0.853**[b] | 0.079 | −0.018 | −0.236 | **−0.794**[b] | 0.224 |
| *Gammaproteobacteria* | 0.255 | −0.239 | −0.467 | 0.333 | 0.139 | 0.491 | −0.576 |
| *Bacteroidetes* | −0.146 | 0.497 | −0.139 | −0.321 | **−0.673**[a] | **−0.709**[a] | −0.430 |
| *Actinobacteria* | 0.182 | **−0.632**[a] | −0.006 | 0.200 | 0.382 | **0.648**[a] | 0.176 |
| *Cyanobacteria* | 0.292 | −0.485 | **0.721**[a] | 0.030 | **0.661**[a] | 0.394 | **0.636**[a] |
| *Epsilonbacteraeota* | −0.219 | −0.080 | −0.273 | −0.200 | −0.564 | −0.176 | −0.515 |
| *Firmicutes* | 0.316 | −0.178 | −0.248 | 0.600 | 0.600 | 0.236 | −0.030 |
| *Patescibacteria* | 0.365 | **0.669**[a] | −0.006 | −0.006 | 0.042 | −0.576 | −0.055 |

Notes:
[a] Correlation is significant at the 0.05 level.
[b] Correlation is significant at the 0.01 level.

*Alphaproteobacteria*) displayed positive correlations with the water temperature ($P < 0.05$ or $P < 0.01$), but were negatively correlated to TP ($P < 0.05$ or $P < 0.01$) (Table 3). Planktonic bacterial community Shannon diversity showed a significant negative correlation with water $NH_4^+$–N and TOC concentrations ($P < 0.05$). Moreover, *Bacteroidetes* displayed a negative correlation with TN and TP ($P < 0.05$), while *Cyanobacteria* was positively correlated to $NH_4^+$–N, TN, and TOC ($P < 0.05$). In addition, water temperature and TP were identified as key drivers determining the abundance of *Actinobacteria* ($P < 0.05$). Water temperature also positively correlated to *Patescibacteria* ($P < 0.05$). The results of RDA analysis revealed that the water environmental factors in the first two RDA axes respectively accounted for 59.1% and 24.3% of the total variance for planktonic bacterial OTU composition (Fig. 6). Rainfall ($F = 7.31$, $P = 0.012$, 499 Monte Carlo permutations), water temperature ($F = 6.93$, $P = 0.006$, 499 Monte Carlo

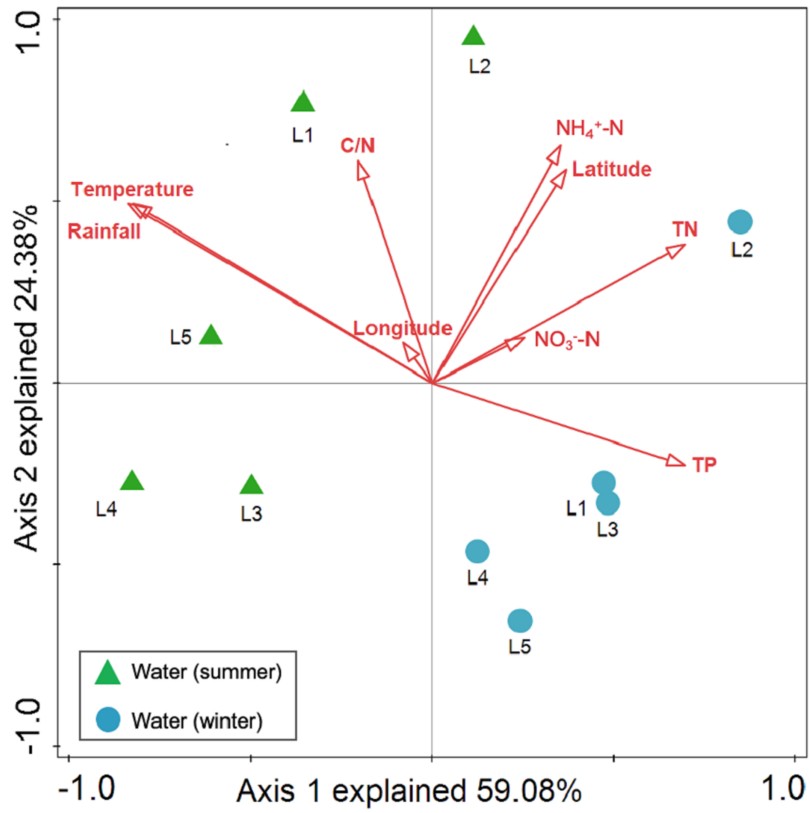

**Figure 6 Redundancy analysis (RDA) for the first twoprincipal dimensions of the relationships between planktonic bacterial OTU composition and water properties.** Green triangle symbols represent the August water samples from sites L1 to L5. Blue circle symbols represent the February water samples from sites L1 to L5.

permutations), TP ($F = 4.05$, $P = 0.028$, 499 Monte Carlo permutations), and TN ($F = 4.00$, $P = 0.022$, 499 Monte Carlo permutations) significantly contributed to the planktonic bacterial population–environment relationship. The VPA indicated that season, nutrients, and geographic distance explained 78.7% of the variation in the planktonic bacterial community (Fig. 7A). Season, nutrients, and geographic distance explained 42.5%, 18.1%, and 13.7% of the total variation, respectively. Therefore, the variation in planktonic bacterial community was mainly determined by season and nutrient condition.

Sediment bacterial Shannon diversity exhibited a positive correlation with sediment $NH_4^+$–N, $NO_3^-$–N, and TN concentrations ($P < 0.05$) (Table 4). *Proteobacteria* was negatively correlated to $NH_4^+$–N and $NO_3^-$–N ($P < 0.05$), while *Deltaproteobacteria* displayed a positive correlation with sediment pH ($P < 0.01$). Moreover, *Alphaproteobacteria* were positively correlated to the water temperature ($P < 0.05$), but negatively correlated to TP and TOC ($P < 0.01$). *Actinobacteria* was positively correlated to $NH_4^+$–N, $NO_3^-$–N, and TN ($P < 0.05$), while *Acidobacteria* showed a negatively correlation with $NH_4^+$–N, $NO_3^-$–N, and TN ($P < 0.05$). *Chloroflexi* was found to be related to sediment pH, TP, and TOC ($P < 0.05$ or $P < 0.01$). Additionally, *Gemmatimonadetes* and *Nitrospirae* were positively correlated to $NH_4^+$–N and $NO_3^-$–N

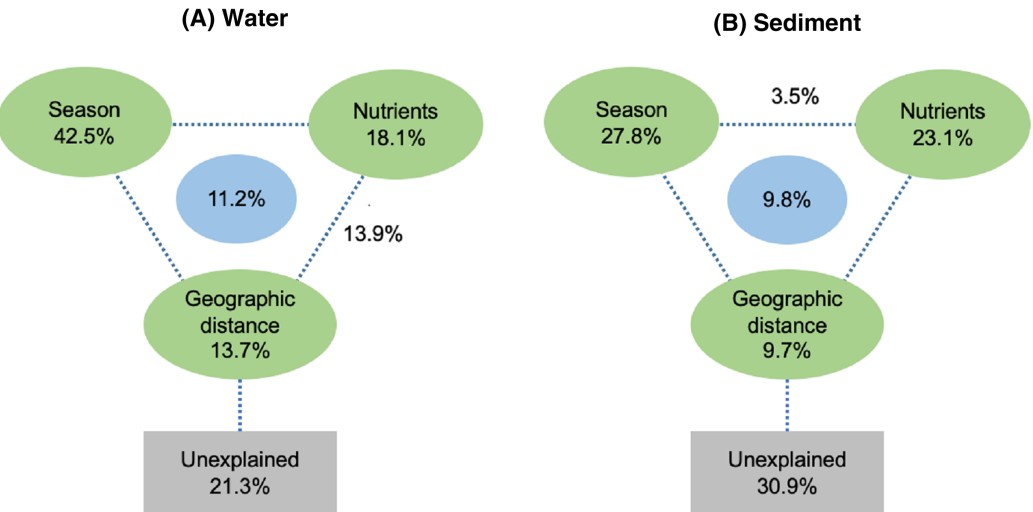

**Figure 7 Variation partitioning analysis (VPA) of the effects of season, nutrients, and geographic distance on planktonic (A) and sediment (B) bacterial communities.**

**Table 4 Spearman rank correlation analysis of sediment environmental factors with the richness and diversity of sediment bacterial community or the proportion of the major sediment bacterial groups. The data in bold indicate significant differences.**

| Parameters | pH | Temperature | $NH_4^+-N$ | $NO_3^--N$ | TN | TP | TOC |
|---|---|---|---|---|---|---|---|
| OTUs | −0.103 | 0.299 | 0.030 | 0.176 | −0.018 | −0.309 | −0.297 |
| Chao1 estimator | −0.212 | 0.079 | 0.055 | 0.321 | 0.006 | −0.030 | −0.006 |
| Shannon index | 0.430 | 0.177 | **0.709**[a] | **0.745**[a] | **0.758**[a] | −0.079 | −0.103 |
| *Proteobacteria* | −0.479 | 0.128 | **−0.770**[a] | **−0.661**[a] | −0.600 | −0.030 | −0.030 |
| *Alphaproteobacteria* | −0.042 | **0.695**[a] | −0.624 | −0.527 | −0.358 | **−0.733**[a] | **−0.709**[a] |
| *Gammaproteobacteria* | −0.503 | −0.28 | −0.467 | −0.491 | −0.394 | 0.248 | 0.236 |
| *Deltaproteobacteria* | **0.770**[b] | 0.488 | 0.455 | 0.515 | 0.539 | −0.236 | −0.164 |
| *Bacteroidetes* | 0.103 | −0.622 | 0.539 | 0.467 | 0.309 | **0.842**[b] | **0.794**[b] |
| *Actinobacteria* | −0.055 | −0.482 | **0.673**[a] | **0.636**[a] | **0.648**[a] | 0.358 | 0.261 |
| *Chloroflexi* | 0.164 | **0.701**[a] | −0.030 | 0.042 | 0.079 | **−0.782**[b] | **−0.758**[a] |
| *Firmicutes* | −0.079 | 0.037 | −0.006 | −0.127 | −0.055 | 0.067 | 0.103 |
| *Acidobacteria* | −0.055 | 0.305 | **−0.733**[a] | **−0.636**[a] | **−0.758**[a] | −0.515 | −0.430 |
| *Gemmatimonadetes* | 0.552 | 0.171 | **0.782**[b] | **0.673**[a] | 0.612 | −0.127 | −0.103 |
| *Nitrospirae* | 0.527 | −0.055 | **0.879**[b] | **0.709**[a] | 0.600 | 0.115 | 0.127 |

Notes:
[a] Correlation is significant at the 0.05 level.
[b] Correlation is significant at the 0.01 level.

($P < 0.05$ or $P < 0.01$), while *Bacteroidetes* showed a positive correlation with sediment TP and TOC concentrations ($P < 0.05$ or $P < 0.01$). RDA analysis for sediment communities showed that sediment physicochemical factors explained 44.5% and 31.4% of the total variation (Fig. 8), respectively. Sediment TP ($F = 3.89$, $P = 0.006$, 499 Monte Carlo permutations), rainfall ($F = 3.38$, $P = 0.016$, 499 Monte Carlo permutations) and

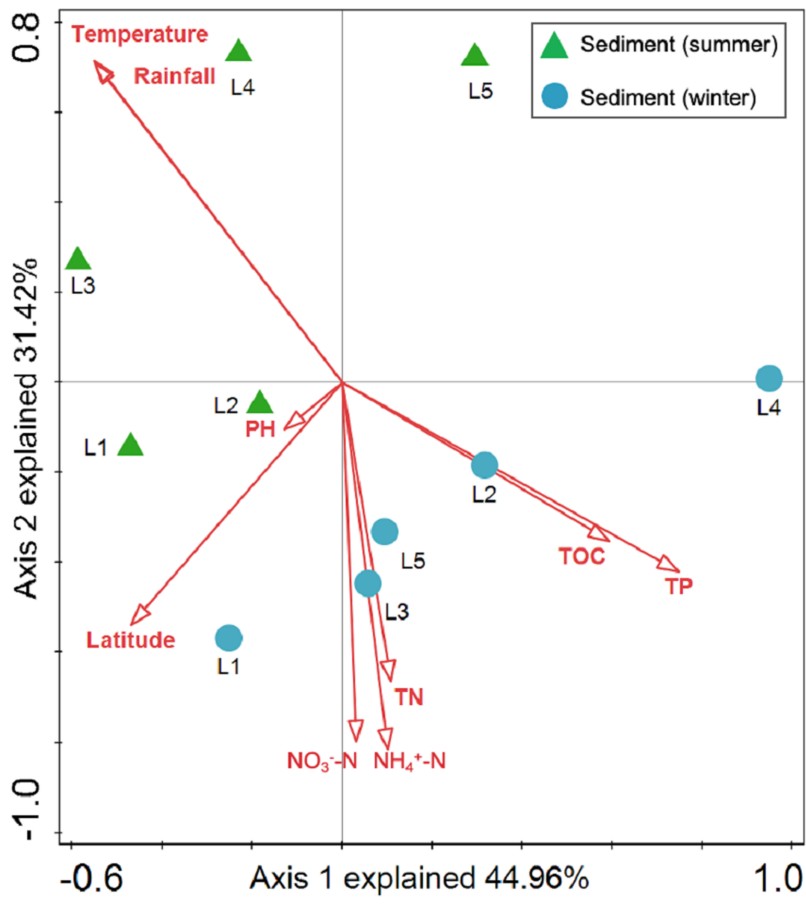

**Figure 8 Redundancy analysis (RDA) for the first two principal dimensions of the relationships between sediment bacterial OTU composition and sediment properties.** Green triangle symbols represent the August sediment samples from sites L1 to L5. Blue circle symbols represent the February sediment samples from sites L1 to L5.

temperature ($F = 3.38$, $P = 0.012$, 499 Monte Carlo permutations) passed the Monte Carlo significance test ($P < 0.05$), suggesting a significant contribution to the sediment bacterial community composition–environment relationship. The VPA showed the season, nutrients, and geographic distance contributed 69.1% of the variation in the sediment bacterial community (Fig. 7B). Season and nutrients explained 54.4% of the variation, indicating a major contribution in shaping the sediment bacterial community.

## DISCUSSION

### Bacterial community richness and diversity in Liangtan river

Numerous previous studies have shown the spatiotemporal variation of bacterial community richness and diversity in aquatic ecosystems (*Adhikari et al., 2019*; *Dai et al., 2016*; *Mai et al., 2018*; *Mao et al., 2019*; *Yang et al., 2019*; *Sheng et al., 2016*; *Zhang et al., 2014*, *2019a*; *Zhu et al., 2019*). However, little is known about the shift of bacterial richness and diversity in given urban rivers. In this study, Illumina Miseq high-throughput sequencing was used to reveal the richness and diversity of planktonic and sediment
bacterial communities in Liangtan river. The Chao1 richness estimators and Shannon diversity index of planktonic bacterial communities were 2,865–3,810 and 4.49–5.20, while those of sediment communities were 4,083–6,015 and 6.40–7.09, respectively. However, lower Chao1 richness estimator but higher Shannon diversity index of both planktonic and sediment bacterial communities were detected in Jiangling river, the mainstream of Liangtan river (*Wang et al., 2018*). Additionally, the Chao1 richness and Shannon diversity of sediment in Liangtan river were markedly higher than those of their corresponding water column, which were in agreement with previous studies in river systems (*Dai et al., 2016*; *Mao et al., 2019*; *Staley et al., 2015*). In this study, a remarkable seasonal fluctuation of planktonic bacterial community richness occurred in Liangtan river, but the trend of seasonal variation of sediment community diversity was not clear. Moreover, significant differences in Shannon diversity of both planktonic and sediment bacterial communities were observed between HS and RI areas in summer ($P < 0.05$), suggesting that the spatial distribution of bacterial communities. The spatial change of bacterial community diversity was dependent on sampling date. *Dai et al. (2016)* indicated a remarkable spatial variation of bacterial Shannon diversity in summer compared to spring, which further supported the results.

Previous literatures suggest that planktonic bacterial community richness or/and diversity in river ecosystems might be influenced by water temperature, pH, and nutrients (*Mao et al., 2019*; *Staley et al., 2015*; *Wang et al., 2016a*, *2016b*). The results of Spearman rank correlation analysis revealed that water temperature and TP were the dominant driving forces for planktonic bacterial community richness, while $NH_4^+$–N and TOC were identified as key drivers determining planktonic bacterial community Shannon diversity. Numerous influential factors on sediment bacterial communities in river systems have been documented previously (*Mao et al., 2019*; *Zhang et al., 2019b*), where bacterial diversity was correlated with nutrients and heavy metals. *Sheng et al. (2016)* indicated that increases in TN and TP may lead to high bacterial richness and diversity. Our study identified the sediment bacterial Shannon diversity was positively correlated to the level of sediment $NH_4^+$–N, $NO_3^-$–N, and TN, yet no determined indicators could well explain the sediment bacterial richness.

## Bacterial community composition in Liangtan river

There is a large degree of variation in bacterial communities in terms of taxonomic composition in aquatic ecosystems (*Adhikari et al., 2019*; *Chen et al., 2018*; *Dai et al., 2016*; *Mai et al., 2018*; *Sun et al., 2011*; *Xia et al., 2014*; *Yang et al., 2019*; *Zhang et al., 2014*; *Zhu et al., 2019*), whereas the characteristics of bacterial communities in given urban rivers remains poorly understood. In this study, the results of PCoA and PerMANOVA revealed strong heterogeneity in the structure of planktonic and sediment bacterial communities, which was supported by previous studies in river systems (*Ibekwe, Ma & Murinda, 2016*; *Liu et al., 2018*; *Staley et al., 2015*; *Wang et al., 2018*). However, a strong overlap (46.7%) in OTUs was identified between water and sediment samples from Liangtan river. The coexisting bacteria accounted for 76.9% of the whole communities in water column, indicating that planktonic bacterial communities was mostly seeded from

sediment bacterial communities. As proposed by *Staley et al. (2015)*, sediment can contribute up to 50% of the community composition in the water column explaining the big overlap. The positive and negative correlation in water and sediment accounted for nearly 50%, indicating a similar co-occurrence patterns that complex ecological relationships (e.g., symbiosis, competition, or predation) in both water and sediment. However, a higher co-occurrence were found in water, which might be attributed to environmental filtering and niche differentiation.

The most abundant phylum in water and sediment of Liangtan river was *Proteobacteria*, which was in line with previous studies in river ecosystems (*Liu et al., 2012*; *Liu et al., 2018*; *Mao et al., 2019*; *Staley et al., 2015*; *Tao et al., 2019*; *Wang et al., 2018*; *Zhang et al., 2019b*). Noticeably, the genera *Pseudomonas*, *Arcobacter*, *Acinetobacter* belonging to phylum *Proteobacteria* were detected in high abundance in water and/or sediment samples. Additionally, taxa displayed higher abundances in the RI area compared to the HS area (Table S3) that may be attributed to the high nutrient sources from industrial and residential sewage. *Newton et al. (2013)* identified three bacterial genera (*Acinetobacter*, *Arcobacter*, and *Trichococcus*) as sewer signatures in urban wastewater. *Yang et al. (2019)* revealed that members of genus *Acinetobacter* comprise of potentially pathogenic as well as sewage indicator bacteria. *Bacteroidetes* was the second largest phylum in the water samples from Liangtan river. Specific species within this phylum are associated with the gut microbiota of many mammals (*Shanks et al., 2011*) and humans (*Candela et al., 2010*). Consequently, Bacteroidetes might be proposed as effective alternative fecal indicators (*Wéry et al., 2008*). In our study, water samples from RI area generally had higher abundance of the potential pathogenic bacteria *Bacteroides* (phylum *Bacteroidetes*) than those from HS area (Table S3). Another indicator of fecal contamination is *Firmicutes*, which is one of the markers for identifying human feces (*Zheng et al., 2009*). The relative abundance of *Firmicutes* was higher in sediment samples than that in water samples. Interestingly, the genera *Bacillus* and *Clostridium* (phylum *Firmicutes*) were found mainly in sediment samples from RI area associated with the presence of untreated fecal sewage.

*Cyanobacteria* was another predominant group of planktonic bacterial communities in Liangtan river. A remarkable increase of *Cyanobacteria* abundance in site L2 compared to that in site L1 was detected which might be associated with hydropower station. Similar observations were documented for the Guadiana river (*Domingues et al., 2012*) and Jiangling river (*Wang et al., 2018*). The impact of dam construction was predicted to increase eutrophication conditions and possibly promote Cyanobacteria blooms (*Wang et al., 2018*). Additionally, the abundance of *Cyanobacteria* in our study was positively correlated to water $NH_4^+$–N, TN, and TOC concentrations, suggesting its potential to serve as an important indicator of river eutrophication. These results suggest environmental dominators drive the potentially responses of bacterial community structure to the domestic sewage-polluted conditions and construction of hydropower station.

The VPA revealed a major contribution of season and nutrients in shaping planktonic and sediment bacterial communities, yet the effect of geographical distance on bacterial community structure was negligent. Environmental factors are more likely to contribute

to variation in bacterial composition than geographical distance at small or intermediate spatial scales (less than 3,000 km; *Martiny et al., 2006*). Hence, environmental factors (season and nutrients) were more important than geographical distance in shaping bacterial community structure in Liangtan River. Specifically, the results of RDA suggested that rainfall, water temperature, TP, and TN between seasons mainly accounted for the shift of the planktonic bacterial community structure, while TP, rainfall and temperature were identified as key drivers determining sediment bacterial community composition. The fluctuations of rainfall and temperature between seasons may mainly account for the seasonal variations in bacterial community structure. As proposed by previous studies (*Haque et al., 2020*; *Staley et al., 2015*), seasonal hydrological regime (i.e., rainfall) directly causes a "wash-out effect" of nutrients and bacterial assemblages. Additionally, nutrients were found to be another key driver explaining bacterial community structure. *Ibekwe, Ma & Murinda (2016)* highlighted that phosphorus and nitrogen are the major factors significantly related to the changes in planktonic bacterial community. *Zhang et al. (2019b)* also stressed the significant roles of TP and temperature on the variation of sediment bacterial community.

## Implications of the research

Sediments within aquatic ecosystems represent a complex matrix, usually providing microorganisms with numerous advantages not found in the water column.

*Perkins et al. (2014)* reported that sediments contain more nutrients than the water column, and sediments can shield bacteria from the effect of sunlight and protect bacteria against predation.

In the current study, the sediment samples clustered distinctively from water samples and displayed a higher bacterial diversity than the water samples, strengthening the fact that sediments may serve as reservoirs of diverse microbial populations and highlighting the need to include sediments in river monitoring. Additionally, the fecal bacteria and candidate pathogens accounted for a higher proportion in RI area, suggesting their potential to serve as important indicators for the health and status of river ecosystems. Nevertheless, more extensive researches should be conducted in our future work, including (i) a long-term investigation (i.e., samples will be collected seasonally during 2 years) of planktonic and sediment bacterial communities in urban rivers polluted by continuous domestic sewage is of paramount importance; (ii) the role of functional microorganisms, especially high abundance of bacterial pathogens, is furthermore elucidated to provide insight into the monitoring and control of contamination in urban rivers.

## CONCLUSIONS

Our study revealed a comprehensive understanding of the integrated biogeography of planktonic and sediment bacterial communities in Liangtan river. Water and sediment habitats differed greatly in bacterial community richness, diversity and structure. Additionally, the relative abundance of potential bacterial pathogens in water and sediment samples from RI area was significantly higher than those from HS area.

Under the joint influence of sewage discharge and hydropower dams, rainfall, temperature and nutrient were identified as the deterministic forces shaping the bacterial communities in the heavily polluted urban river. Our study offers further reference for the environmental management of Liangtan River. It also provides typical data for the comparison with other urban rivers, and thus contributes to the increasing knowledge of microbial ecology in contaminated urban rivers.

### Funding

This work was supported by the Performance Incentive for Scientific Research Institutions of Chongqing (cstc2018jxjl20018; cstc2018jxjl20012; cstc2018jszx-zdyfxmX0021). The funders had no role in study design, data collection and analysis, decision to publish, or preparation of the manuscript.

### Grant Disclosures

The following grant information was disclosed by the authors:
Performance Incentive for Scientific Research Institutions of Chongqing:
cstc2018jxjl20018, cstc2018jxjl20012 and cstc2018jszx-zdyfxmX0021.

### Competing Interests

The authors declare that they have no competing interests.

### Author Contributions

- Heqing Huang conceived and designed the experiments, analyzed the data, prepared figures and/or tables, authored or reviewed drafts of the paper, and approved the final draft.
- Jianhui Liu performed the experiments, authored or reviewed drafts of the paper, and approved the final draft.
- Fanghui Zhang performed the experiments, prepared figures and/or tables, and approved the final draft.
- Kangwen Zhu performed the experiments, prepared figures and/or tables, and approved the final draft.
- Chunhua Yang performed the experiments, analyzed the data, authored or reviewed drafts of the paper, and approved the final draft.
- Qiujie Xiang performed the experiments, authored or reviewed drafts of the paper, and approved the final draft.
- Bo Lei conceived and designed the experiments, analyzed the data, prepared figures and/or tables, authored or reviewed drafts of the paper, and approved the final draft.

### DNA Deposition

The following information was supplied regarding the deposition of DNA sequences:
The obtained raw reads are available at the NCBI Short-Read Archive: SRP248929.
## Data Availability

The raw measurements are available in the Supplemental Files.

## Supplemental Information

Supplemental information for this article can be found online at http://dx.doi.org/10.7717/peerj.10866#supplemental-information.

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
