# Peer review of "Characteristics of planktonic and sediment bacterial communities in a heavily polluted urban river"

_PeerJ, doi:10.7717/peerj.10866_

## Round 0.1 · original submission · Major Revisions

Dear Dr. Huang,

After reading your MS, I have a similar view to the ones expressed by both reviewers. Please, consider providing more background information on the sampling sites and the differences among them, including the distance as pointed by reviewer 2 Basic Reporting topic 5. For example, in lines 87 - 89, you mention this river water quality has always been lower than 5 based on national standards. But does this index score the river just in its whole, not in sections? I would recommend to include the map (Figure S1) in the main MS. Moreover, the sampling site codes in the figures and tables could be more intuitive and must be explained in every figure or table caption. Please, provide a more explained caption for figures and tables in the main MS. The data provided in your MS is novel and relevant, but I consider the questions raised during this revision process need to be clarified and that our comments can potentially strengthen your MS. Looking forward to a revised version.

·

Basic reporting

The paper is clear, but the use of English is a bit poor.

The figures, tables and raw data are clear and well presented, with professional structure.

The background / context is poor.

Experimental design

Sampling design is not justified, neither the expected differences among sites.
The ecological differences between planktonic and benthic communities are not well defined, and consequently, the scientific question addressed does not have the appropriate conceptual framework. The importance of nutrient enrichment is poorly developed.

Validity of the findings

The results are valid and novel.

Additional comments

The paper is clear, but the use of English is a bit poor.

The figures, tables and raw data are clear and well presented, with professional structure.

The background / context is poor. There ecological differences between planktonic and benthic communities are not well defined, and consequently, the scientific question addressed does not have the appropriate conceptual framework.
The importance of nutrient enrichment is poorly developed. Sampling design is not justified, neither the expected differences among sites.

·

Basic reporting

1) The English language, grammar style and usage throughout the manuscript was good.

2) Sufficient background, with references, was provided in the Introduction section. I have the following suggestions:
p. 8 line 62-73: The transition between the first (surrounding terrestrial environment) and second (dams) part of the paragraph is not fluent. An extra sentence(s) is necessary to inform the reader how dams play a role in aquatic ecosystem.

3) Raw data for environmental parameters measured were not supplied. It will be helpful to add a table with the values of environmental parameters so the reader can trace back any patterns/variations to the raw data.

4) Some of the figures can be enhanced. For example, water and sediment samples were plotted on two separate RDA plots. It will be useful to make one RDA plot for all samples. It will be easier for the reader to see/determine which environmental parameters influence which samples.

5) The aim was to investigate “spatiotemporal” dynamics, however, only temporal trends were investigated and discussed. Distances between sampling points were not reported, so it is not clear whether spatial trends in community dynamics were possible/observed.

Experimental design

1) It is not clear if community data were log/square root transformed prior to beta diversity calculations and RDA analysis.

2) It is not clear if environmental data standardized (z-score) prior to RDA analysis.

3) I recommend that the authors perform principal component analysis (PCA) to determine the most important gradients in environmental data.

4) It will be useful if the authors can perform additional statistical analysis on the alpha and beta diversity metrics. For example, ANOVA, Kruskal-Wallis or Wilcoxon tests can be used on alpha diversity metrics to determine statistical differences between samples. For beta diversity, PERMANOVA (adonis/adonis2) can be used to determine if there is a significant difference in the centroid & dispersion of upstream and downstream communities. This can be supplemented with betadispersivity (betadisper) tests.

5) Were Spearman rank correlations performed on absolute or relative abundances? It is better to use absolute abundance for this purpose because relative abundance data suffers from apparent correlations. That being said, the paper by Quinn et al. 2017 (https://www.nature.com/articles/s41598-017-16520-0) is very helpful to understand compositional data and correlations.

6) Did the authors perform weighted or unweighted Unifrac? Why did they use UPGMA clustering to visualize beta diversity? I believe visualization of beta diversity will be better with a PCoA or NMDS plot.

7) I commend the authors for applying DCA to determine which ordination analysis fits their data, very few papers report on this. However, the description of the process can be shortened (e.g. Redundancy analysis (RDA) was performed (Lepš and Šmilauer, 2003) based on the DCA results.)

8) I have the following suggestions for in-text corrections:
p. 10 line 107: to retain microbial cells from river water
p. 10 line 109: according to manufacturer instructions.
p. 10 line 110: using the primer set
p. 10 line 117: further processed as previously described (Caporaso et al., 2010). Chimeric reads were discarded using UCHIME (Edgar et al., 2011).

Validity of the findings

1) Alpha diversity indices are presented in text. I recommend that indices are shown as box plots (with median values and interquartile ranges indicated on plots); statistical differences between groups can also be indicated on plots. Visualization of indices will make it easier to follow spatiotemporal trends.

2) Authors refer to “significance/significant” results in text (p. 12 line 155 & 167) but they fail to mention if statistical analyses were performed on alpha or beta diversity metrics in the Materials & Methods sections. If such analyses were performed it have to be clearly stated.

3) The study made conclusions only on the temporal aspect of bacterioplankton and sediment communities. The manuscript can be improved by not only looking at which environmental parameters influence communities, but also which anthropogenic activities impacted their diversity and distribution. This can be achieved by variation partitioning analysis or a similar type of analysis. It will also be good to infer relationships/interactions between OTUs within communities and/or between communities (e.g. water and sediment). This will provide a deeper insight into community dynamics but also strengthen the paper.

---

## Round 0.2 · Major Revisions

Dear Authors,

I support the reviewer's comments on both the attendance to most of the previous suggestions and the need for new & previous corrections/suggestions.

Please, read the manuscript carefully and make the corrections detailed by the reviewer (and possibility others).

In addition, please standardise and clarify the samples names/codes throughout the manuscript, including the figures.

The manuscript discussion would be strengthened if the questions raised by the reviewer in the section "validity of the findings" were addressed.

Looking forward a revised version.

King regards,

·

Basic reporting

1) The manuscript is written in English but technical grammar can be improved.
2) The authors updated the Introduction section to provide additional background/context with sufficient references.
3) The authors have uploaded the environmental data as previously requested. Figures are relevant to the content of the manuscript and of sufficient resolution. However, I do have comments regarding the following figures:

• In Figures 3-6 & 8 different sample names (SFe1-5, SAu1-5) are provided without clarification in the Materials & Methods section, whereas in the latter section only sampling sites (L1 – L5) are mentioned. This might be confusing for some readers and needs to be clarified.
• There’s an error in legend for Figure 3. The colouring for ‘Sediment (winter)’ should be blue and not green
• Figure 6: Instead of using latitude and longitude in the RDA, the authors can use distance between sampling points as a variable.

4) The results presented in the manuscript are relevant to the aim.

Experimental design

1) The aim of this study was clearly stated in the Introduction section
2) The authors described the methods with much greater detail compared to the first manuscript. I want to point out the following aspects:
• Air temp for winter was warmer than for summer. Maybe the two temperatures were switched?
• The authors calculated statistics between sample groups (SFe1-5, SAu1-5) but failed to mention the groups in the Materials & Methods section. This can cause confusion with readers as the authors only mentioned the sites (L1 – L5).
• The authors provided some seasonal information for the sampling sites, additional information such as rainfall can be extremely helpful. I have noticed that nutrient levels during winter was higher than in summer that might be linked to for e.g. rainfall events and river hydraulics. Also, when nutrients are high it might select for specific taxa that can influence diversity.
• Although the authors stated in their rebuttal letter that species data were log transformed and environmental data were z-score standardized prior to beta diversity calculations, it is important to mention it in the text as well (which is currently missing).
• It is also not clear if the authors tested for multicollinearity (VIF) and significance (anova) of environmental parameters before running the final RDA model.

Validity of the findings

1) Although the results are interesting and additional statistical analyses were performed, I believe that potential functionality and co-occurrence networks between water and sediment bacterial communities can provide some insight into how the two communities coincide and function. Such results will greatly improve the manuscript.
2) The Discussion and Conclusion sections will also benefit if the authors can discuss the significance of the results and not only compare their results to previous studies. For example, the authors can look into the following points:
• What do the results imply about the health and status of the river and for use by the community?
• Is it possible to infer long-term consequences on bacterial communities (water column and sediment) as well as river health if pollution continues?
• Can the results be used for any monitoring purposes?
• What are the limitations of this study and plans for future improvements?

Additional comments

No comment

---

## Round 0.3 · Minor Revisions

Dear Dr. Lei,

I congratulate you and your collaborators for the improvements made on the MS.

There are, however, some issues that need to be included, explained or elaborated. Please refer to the reviewers comments. I highlight the importance to include the method used to constructed the co-occurrence network or to reassess the value it added to the work.

Looking forward a revised version.

Kind regards,

·

Basic reporting

1) The manuscript is written in English and the technical grammar was improved.
2) The authors updated the Introduction section as suggested to provide additional background/context with sufficient references.
3) The authors amended the Figures which are relevant to the content of the manuscript and of sufficient resolution.
4) The results presented in the manuscript are relevant to the aim.

Experimental design

1) The aim of this study was clearly stated in the Introduction section
2) The authors described the methods with much greater detail compared to the second version. I want to point out the following aspects:

• The method used to construct the co-occurrence network was not mentioned in the Materials & Methods section. Did the authors used correlation analysis, MENA etc. to construct the network? Also, what are the network properties?
• Did the authors include Rainfall as an environmental parameter in correlation and RDA analyses? There was a big difference between summer and winter rainfall and one will expect it to significantly influence communities and taxa.

Validity of the findings

1) I appreciate the fact that the authors tried to include a co-occurrence network. However, the network does not add any value; it says nothing about the interactions between water and sediment species. It only states the percentage of OTUs shared between water and sediment samples which is more like a Venn diagram. I suggest that the authors reassess the construction and interpretation of co-occurrence networks and the value it can add to the study.
2) As indicated by the authors, winter and summer rainfall differed markedly that may also explain the high abundance of some taxa and correlations between nutrients and taxa. The authors explained/discussed the influence of other (seasonal) environmental parameters on bacterial communities but failed to discuss the impact of rainfall, which is one of the most important seasonal parameters. As suggested previously, rainfall events impact river hydraulics and can even cause a “wash-out effect” of nutrients and microbial communities. Also, when nutrients are high (e.g. winter) it might select for specific taxa that can influence diversity.
3) The Conclusion section is very short and can be elaborated to include more detail.

---

## Round 0.4 · Minor Revisions

Dear Dr. Huang,

Thank you for submitting a revised version of your MS taking many of the reviewer's comments into account.

Please, as noted by the reviewer, please describe in more details how the data were prepared for the analysis with MENA, as well as your interpretation regarding the 90% of negative correlations.

Kind regards,
Thiago

·

Basic reporting

Basic reporting
1) The manuscript is written in English and the technical grammar was improved.
2) The authors updated the Introduction section as suggested to provide additional background/context with sufficient references.
3) The authors amended the Figures which are relevant to the content of the manuscript and of sufficient resolution.
4) The results presented in the manuscript are relevant to the aim.

Experimental design

Experimental design
1) The aim of this study was clearly stated in the Introduction section
2) The authors gave close attention to the reviewer’s suggestions and described the methods with much greater detail compared to the previous versions. I want to point out the following aspects:
• MENA was used to construct the co-occurrence network. However, no details were provided for data preparation (did the authors use the entire dataset or a filtered dataset; was logarithm used or not; which similarity matrix was used for correlation analysis) and network properties. These details are important to readers for network reconstruction purposes.

Validity of the findings

Validity of the Findings
1) I appreciate the fact that the authors tried to include a co-occurrence network. However, I find it difficult to believe that water bacterial communities had more than 90% of negative correlations. Obviously, there will be competition and predation among species, but the majority of interactions will not consist of these factors. It would make more sense to have such a high percentage of positive correlations. Although the current network displays positive/negative interactions, I suggest that the authors upload an additional metadata table into Cytoscape where OTUs are colored according to phylum level. Observing positive/negative correlations between phyla in water and sediment communities will add great value to the study.
2) The Conclusion section is very short and can be elaborated to include more detail.

---

## Round 0.5 · accepted · Accept

Nice work and revision process. Congratulations.